# Embracing the Collective: Challenges and Opportunities in Teaching and Teacher Education in the Wake of COVID-19

## Noah Borrero

School of Education, University of San Francisco, San Francisco, CA 94117, USA; neborrero@usfca.edu

**Abstract:** Ongoing impacts of the COVID-19 pandemic highlight structural inequities in our educational systems and force educators to ask prominent questions about the role of school as conditions continue to shift. This paper explores challenges and opportunities for moving forward in the fields of teaching and teacher education. Through presenting a framework of education for liberation, sociocultural learning theory, and fostering cultural assets, the value of live, in-person teaching is highlighted. Embracing this moment in history through honoring the lived experiences of students and striving for more collective approaches to teaching and learning are discussed as future possibilities in education.

**Keywords:** education; teaching; teacher education; learning; schooling; sociocultural learning theory

## 1. Introduction

The COVID-19 pandemic continues to impact educational systems around the globe. In the United States (U.S.), where free, public education is compulsory for students ranging from ages 6–18, the challenges associated with forced school closures in March 2020 linger. While nearly all public school districts across the country have returned to in-person instruction, the jolt that all of us (students, teachers, parents, administers, etc.) felt, shook the system in seismic ways. Namely, the abrupt transition to online learning cracked the foundation of learning and teaching in schools—social interaction. Without live, in-person interaction and connection, the very core of education was instantly shifted, and it forced us to ask significant questions about our educational system. Among the many questions being asked, the importance of school came into focus: How can schools best educate students given the shifting conditions initiated by the COVID-19 pandemic?

I am a teacher—formerly a middle- and high-school teacher and now a teacher educator—and I feel that the past three years have created a sea change in the profession. Following their forced closure in the spring of 2020, schools are not the same, and the challenges and opportunities that continue to arise as a part of the ongoing COVID-19 pandemic are constantly reshaping the ways we think about education and its role in our society. In this paper, I attempt to explore some of the ongoing impacts of the COVID-19 pandemic on teachers and teacher education as a way to identify existing challenges and explore future possibilities. Given that the majority of my teaching experiences are based in the U.S., my analysis is focused on the public education system here, yet the global impacts of the pandemic clearly cross borders. As such, I provide a conceptual framework for trying to understand the ongoing challenges and opportunities afforded by the pandemic as a way to think about starting points for how teaching and teacher education can move forward by looking back (Bailey et al. 2022). How can we as teachers learn from the challenges of the COVID-19 pandemic to create more equitable educational opportunities for more students?

## 2. Theoretical Framework

The COVID-19 pandemic continues to highlight structural inequities in our educational systems (Roy 2020). In K-12 public school classrooms in the U.S. these inequities are

most prominent in the lack of consistent, quality educational opportunities for working class students of color (Kozol 2012). The school system prioritizes and proliferates white, middle-class values and beliefs, and as such, reproduces structural racism (Ayers et al. 2018; Delpit 1988). The connections between the theoretical constructs that are outlined below— education for liberation, sociocultural learning theory, and fostering cultural assets—are discussed to provide a framework for understanding the ways in which teaching and teacher education can address some of these inequities moving forward. When considered under the spotlight of the COVID-19 pandemic and the lack of educational opportunities afforded to students of color as compared to their white classmates (Howard 2019), this theoretical framework strives to form a pathway towards honoring the cultural strengths of students of color as a central pillar in the future of meaningful education.

## 2.1. Education for Liberation

Freire's (1970) approach to education centers lived experiences as the foundations of learning. Freire's concept of "reading the word to read the world" expresses the idea that literacy—something so vital in contemporary school settings—is a call for social action. Reading is not just about decoding information on a piece of paper, it is about applying this information to lived experiences. In this way, for Freire's students—and for many students from marginalized communities in classrooms today—reading is a tool for liberation. At the systemic level, school is a place of learning that provides such tools through dialogue and reciprocation by student and teacher (Zavala and Henning 2021). In Freire's framework, much of the goal of education, then, becomes the dismantling of oppression via those that are most oppressed liberating themselves and their oppressors.

In the current context, specific forces associated with the COVID-19 pandemic and the forced school closures of 2020 highlight ways in which the schooling system further oppresses students from communities that are historically targeted. For example, the transition to online learning in March 2020 left school districts struggling to address issues like computer availability and internet access. Yet, for many students and families in working class communities, deeper needs like shelter, food, and health care were vital (Yeh et al. 2022). The entire idea of "essential" workers and who was even afforded the luxury to shelter in place during the initial weeks of the pandemic reflect classist and racist notions that were not discussed nor addressed by schools (James 2021). The result was (and continues to be) further suffering by those who have the least (Chen et al. 2022). As a concept, education for liberation urges us as teachers to think about how students' lived experiences must be at the center of teaching and learning.

## 2.2. Sociocultural Learning Theory

Complementing this approach to systemic change through education is the core belief that social interaction matters. As citizens and learners, we all navigate multiple cultural contexts daily, and these places and spaces are important to how we interpret the world around us and how we construct our identities (Nasir et al. 2021; Paris 2010). Theoretically, social interaction is pivotal to learning because it is through the guidance and mentorship of others that we comprehend our surroundings (Vygotsky 1970). Practically, we all know this to be true because we are continually learning from those who came before us—in our family, in our community, in our field of study, etc. Interdisciplinary approaches to and applications of sociocultural learning theory reinforce the vital role of culture as something that is fluid, dynamic, and learned (Yeh et al. 2021). Thus, we are continually co-constructing learning and teaching with those closest to us.

The forced school closures during the early months of the COVID-19 pandemic stripped away much of this essential component of education—learning with others across multiple cultural contexts. As a result, we all suffered. The spring of 2020 had a devastating global impact—particularly highlighted by unprecedented loss of life. Again, those with the fewest resources suffered the most, as the cumulative effects of racism resulted in disproportionate deaths from COVID-19 among working class communities of color (Fortuna

et al. 2020). In the context of education, this suffering was exacerbated by school closures as young people did not have access to many of the places and people that could have helped support them through such difficult times (Yeh et al. 2022). Not only were they isolated from their peers and teachers, but they did not have access to counselors, nurses, coaches, and other vital social connections.

*2.3. Fostering Cultural Assets*

As a theoretical and pedagogical approach, fostering cultural assets directly counters the racist and hegemonic assumptions that normalize white, middle-class values in education (Camangian 2021). Honoring the intersectional cultural experiences of students of color as foundations of meaningful learning serves as a way to dismantle the deficit-laden mindset in education that exonerates whiteness (Anzaldúa 2021; Crenshaw 2019; Flores 2016). Cultural assets are more than strengths that can be applied to classroom learning; they are interconnected living and breathing actions and traits that showcase the community cultural wealth (Yosso 2005) and funds of knowledge (Moll et al. 1992) that communities possess and share (Borrero and Yeh 2016). In fact, the intergenerational and collectivist cultural priorities of many communities of color in the U.S. (and around the globe) offer insights into the ways in which culture is something we all contribute to and need—cultural assets do not function in isolation.

While the educational impacts of the COVID-19 pandemic have certainly been devastating for many K-12 students of color (James 2021), fostering cultural assets is an approach that may offer some hope for a new vision of classroom teaching and learning (Borrero and Yeh 2021). As in-person classroom instruction continues to evolve after the shock of forced remote learning, there exists potential for reimagining the goals of day-to-day classroom learning. The pandemic has forced all of us as teachers to appreciate the social interaction that is at the core of our profession, and to strive to live in the present—to embrace the vitality of the classroom (Borrero and Yeh 2021; Roy 2020). With this can come a sense of rejuvenation and perhaps a willingness to unlearn (Camangian and Cariaga 2021; Howard and Milner 2021) some of the pre-determined and assumed practices that perpetuate the status quo of "doing school" (Pope 2001). If we can create spaces for students to share about their own experiences living through a global pandemic, there is tremendous potential for cultural assets to emerge as platforms for classroom dialogue, learning, and community-building.

**3. Methods**

I do not present empirical data in this paper. Instead, my attempt is bridge the theoretical framework above to the practice of education through analyzing some of the impacts of the COVID-19 pandemic. Alongside literature from the field of education, my analysis is rooted in my experiences as a mixed-race student in K-12 public schools, my years as a middle- and high-school public school teacher, and my work as a professor of teacher education. As a social sciences scholar, this approach to research attempts to build upon foundations of critical ethnography (Boylorn and Orbe 2020; Ohito 2019), testimonio (Sanchez 2009), and community cultural wealth (Yosso 2005). As a way of highlighting the role of shared cultural experiences that are central to my own understandings of education, critical ethnography (Boylorn and Orbe 2020) provides a foundation for disrupting monolithic assumptions about research that prioritize objectivity. In an attempt to better-understand new possibilities for education, I place myself within this research and attempt to express my own concerns, self-consciousness, and hopes for an improved pathway moving forward. As such, I try to draw from testimonio (Sanchez 2009) and the idea that our experiences matter—particularly when thinking about shifting hierarchical power relations in research and in teaching. Community cultural wealth (Yosso 2005) helps situate this type of counternarrative in a social and psychological framework that values alternative forms of strengths that are rooted in collectivist cultural experiences.

## 4. Confronting Systemic Challenges

The forces working against young people of color in our educational system are powerful and constant; and, there is no doubt that the continued uncertainty, fear, anxiety, and loss caused by the COVID-19 pandemic reinforce and intensify some of the structural injustices that plague our educational system (Yamauchi et al. 2022). In these ways, the challenges posed by the pandemic are both old and new, perennial and urgent.

### 4.1. Long-Standing Educational Inequities

Perhaps the greatest test facing us as teachers is the challenge to embrace this moment—even though we are all suffering. 2020 is a year that we will never forget, and the ideas of recognizing and reflecting on the ways that the COVID-19 pandemic has forever changed all of us are constant and necessary; this process is also vital for our students. Without embracing this time as one that has forever changed our profession and schooling as we know it, we run the risk of simply returning to teaching and learning like we have always done them; this approach does not work—especially for many working-class students of color in our schools (Kumashiro 2020; Hannegan-Martinez 2019).

From this perspective, a core challenge for us as teachers and teacher educators lies within ourselves and our own expectations. Do we have the humility to examine our own complicitness in the promotion of the educational inequities that plague our system? Can we change our own perspectives and make room for new approaches and different voices (Roy 2020)? The goals of a public education system include providing meaningful educational opportunities for all students. Can we acknowledge that our current system does not do this and that we are part of the problem? If we can begin to answer yes to some of these questions, I believe that we can begin to address some of the challenges of dismantling the structural inequities of our schools and classrooms. The pandemic continues to shine a spotlight on these foundational challenges embedded in our system, and there is no better time than now to come together to try to address them.

### 4.2. Stressors of Remote Learning

Undergoing change is a challenging process. By nature, change forces us into new and often times uncomfortable behaviors. Given the significant and constant changes to schooling that the COVID-19 pandemic imposes, a major challenge for us as teachers in this moment involves managing the stress associated with the changing world around us. As teachers, one of these shifts was initiated during the forced closure of schools in March 2020 and the transition to remote instruction. While the vast majority of public K-12 schools are currently operating through in-person instruction, the impact of the abrupt switch to online learning and its continued prominence in the larger educational context lingers. Remote learning continues to loom as a powerful force in teaching and teacher education (Goudeau et al. 2021).

Given the theoretical framework outlined above and my own personal convictions as a life-long teacher and learner, I feel that the threat of remote learning poses distinct and potent challenges to teachers. Live, in-person social interaction matters for meaningful learning and the idea that a classroom session can be made up, substituted for, or happen via Zoom is erroneous. While we might be able to see one another on a screen, an online meeting is inferior to live, in-the-flesh dialogue and discourse. Particularly for young people, the socialization of schooling is vital. And, for us as teachers, it is what attracted us to the profession and what sustains our passion for impacting the future of society. So, a formidable challenge as the COVID-19 pandemic lingers is continuing to value and fight for in-person instruction—at all levels. This is difficult work because there is no doubt that remote learning options can be more convenient, a more efficient use of time, and, in some cases, better for particular students. Yet, we cannot slide down the slippery slope of letting these factors obscure our vision for meaningful in-person teaching and learning. This is not just about advocating for K-12 schools to remain in-person, this is also about modeling behaviors for young people, continuing to hold meetings and professional development

sessions in-person, and actively participating in an educational system that centers social interaction.

## 5. Envisioning Possibilities

Compounding the challenges discussed above is the ongoing uncertainty of the pandemic—we simply do not know how it will continue to impact teaching and teacher education. Within this realm of the unknown is perhaps where some of the greatest opportunities for change and transformation reside. We know that our educational system is deeply flawed (Love 2019) and the impacts of the COVID-19 pandemic have highlighted some of the historical and structural injustices at the core of schooling (Picower 2021). This combination of a broken system and an unpredictable future makes for an opportune moment in the history of education.

### 5.1. Educational Reset

Now is the time to strive for monumental changes in our public education system in the U.S. Gloria Ladson-Billings (2021) writes about this moment as an opportunity to reset the system. It is time to focus on dismantling the racism and dehumanization that many students face throughout their schooling. This includes upending the culturally-biased standardized tests that determine educational pathways for students (Borrero et al. 2019), abandoning archaic curricula that center whiteness as the foundation for meaningful knowledge (Howard 2019), and ending school policies and procedures that criminalize students for resisting cultural assimilation as a requirement for academic success (Hines and Wilmot 2018). Further, this moment provides us the opportunity to question some of the most elemental features of the institution of school—class size, duration of the school day, physical layout of a classroom, and structural hierarchies of school personnel (Yeh et al. 2022). If we are willing to interrogate some of these structures while asking the larger question—how can schools best educate students in this moment?—opportunities can arise to make big changes.

This idea of an educational reset is neither simple nor comfortable. As a teacher myself, I have spent my entire life going to school. I care deeply about my students, my profession, my colleagues, and the future of our public school system (in fact, I have spent much of my adult life defending the pivotal role that our K-12 schools must play in the vitality of democracy), yet I know that deep changes need to happen. If the system is going to change, I need to change. The discomfort of this moment in history—in large part instigated by and embedded in the COVID-19 pandemic—is something that I need to embrace. Perhaps we as teachers are living in the proverbial "teaching moment" and we need to relinquish our control and desires to provide answers and simply try to live in these times and know that the challenges and opportunities surrounding us are beyond our control and cannot be answered right now. Perhaps being with our students at school as the pandemic continues to alter our daily realities is enough.

### 5.2. Intersectionality and Collective Cultural Assets

There is no doubt that young people are going to emerge to lead us into the future—a future beyond the COVID-19 pandemic. In education, this future must be one in which young people's cultural experiences—their traditions, their histories, their languages, and their stories—are a central part of their learning at school. Young people should never be forced to surrender their identities to gain new knowledge and experiences (Ayers 2019). Students—in every school—need to know and feel that their full selves belong and have value (Valenzuela 1999). Part of this involves dismantling monolithic assumptions about students, their lived experiences, and their cultural identities. Embracing traditions (Anzaldúa 2021) and contemporary conceptualizations (Crenshaw 2023) of intersectionality as a framework and pedagogy for honoring and elevating the voices of students of color is a vital step forward. Further, acknowledging and fostering students' cultural assets as a starting point for all learning and teaching can begin to pave a pathway towards more

equitable access and meaningful learning in the classroom. Pedagogy and curriculum are most impactful when students see a direct application to and impact on their lives (Delpit 1988). The most important element in any classroom is the students themselves.

Applying pedagogical approaches that amplify students' cultural assets affords tremendous possibilities for substantive change if we can unlearn our own assumptions about what school is supposed to be and listen to young people. The COVID-19 pandemic has provided us a glaring example of how we need one another during tremendous suffering. Young people continually share that the thing most debilitating about the pandemic (especially the initial months of lockdown) is the isolation and sadness of not being with peers. This elemental foundation of school—social interaction—is a profound cultural strength that many of us take for granted. When it was taken away in March 2020, we were devastated, and arguably, we are still feeling the impacts of this loss. We need one another. Yet, when we look at some of the very foundations of schooling—testing, grades, matriculation, and class standing—we promote and reproduce individualistic and solitary outcomes. Academic success is almost always seen as an independent pursuit. In many ways, online learning exacerbated (and continues to exacerbate) this reality of our education system. This pillar of our schooling system needs to be knocked down. We need to listen to young people, learn from their collective desires, embrace their collectivist cultural experiences, and make schools more community-oriented spaces. Many students of color from working-class communities have deep knowledge of and connections to collectivist cultural experiences and traditions that we can all learn from and we need to make these experiences foundational in schooling.

## 6. Conclusions and Prospects

The COVID-19 pandemic grinds on, and as it does, we are left to wonder how, when, and if it will end. While this type of questioning is understandable, perhaps an alternative goal is needed. That is, instead of yearning for a post-pandemic reality, the challenge facing us involves embracing the tensions and tumultuousness of this moment in history. The suffering inflicted by COVID-19 has much to teach us, and remaining present during times of prolonged struggle is a worthy pursuit. The isolation and loneliness prompted by the initial months of the pandemic linger, and they compound the challenges of trying to live in the present.

In the field of education, the tumult of the pandemic forced major questions about possibilities for remote instruction, the value of teaching, and the overall role of school during these shifting conditions. After several months of online instruction in many countries, the return to physical classroom learning has triumphed, and the vital role of live, in-person social interaction shines brightly in the spotlight of the pandemic. Contemporary learning theory (Nasir et al. 2021) and our own lived experiences tell us that we learn best in the presence of others. As teachers, great challenges and opportunities exist in this moment—how can we remain present with our students during times of great uncertainty and loss? How can we challenge our own assumptions and biases about schooling to make room for new and different possibilities in the future? How can we hold onto our belief in and commitment to public education while also seeking to transform the system itself?

There are no easy answers to these questions. One possible starting point towards a more equitable education system involves honoring the importance of social interaction in learning and making space for young people to learn from and with one another through dialogue about their experiences during the pandemic. Teaching in this moment is difficult because there are no answers—we do not know what the future holds. We can, however, sit alongside our students in community and embrace a part of school that is a true gift—each other. Collectivity, connectivity, and collaboration are central to meaningful teaching and learning, and the COVID-19 pandemic demonstrates that we must support and strengthen these pillars of education. If we can do so, we can—collectively—begin the process of embracing and enacting new educational reform, policy, and practice that honor and sustain the intersectional and collective cultural assets of students and their communities.

**Funding:** This research received no external funding.

**Institutional Review Board Statement:** Not applicable.

**Informed Consent Statement:** Not applicable.

**Data Availability Statement:** Not applicable.

**Conflicts of Interest:** The author declares no conflict of interest.

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
