# Peer review of "Embracing the Collective: Challenges and Opportunities in Teaching and Teacher Education in the Wake of COVID-19"

_socsci, doi:10.3390/socsci12030194_

Round 1

Reviewer 1 Report

The work presented is a reflection on the effects of the pandemic on education.

As a reflection, I find it interesting and it remains in the field of theory.

The contribution of the work is limited, since it would be the starting point for the proposal of teacher training in order to implement the suggestions made in the classroom.

The structure of the work is correct and the bibliography used is current and relevant.

Reviewer 2 Report

First, I congratulate the author on her boldness to write this paper as a teacher leaning on her lived experiences and how she perceives the inconsistencies across her career and yet as exacerbated during the Covid 19 pandemic. Being not in an academic space, I command her on her ideas, philosophy, writing and use of a conceptual frame and relevant literature, from a sharp and critical edge. 

I would recommend however, a section on research methodology to make this paper stronger, although the author brushes a strong section on her positionality and lived experiences across the paper.  Could consider critical autoethnography. 

Theoretical lenses - few used such as Freire's critical pedagogy, social learning theory, however we need to see how this all works out with the idea of collectivism, cultural learning models for more access, equity and inclusion.  may be a figure to show where the author sits and explains all this in relation to the US context and the challenges of students of Color.

As the paper strongly shows a dilemma in and with students of colour and their disadvantages - I wonder if the author can read and look into the ideas of an intersectionality by Kimberle Crenshaw and others and use the intersections race, class, culture and identity to discuss the issues of reforms of education system and why? collectivism, cultural assets, educational reset ... will sit well here?

In few places, there needs to be some more referencing done, minor though, please go through where I have highlighted in yellow.

Conclusions need to stress on the idea of educational reform, and why collective cultural asset and learning is needed. The COVID 19 online learning seems to just a trigger to create a wave or a ripple effect to critically reflect and respond to a longstanding systemic educational stance in this particular context of this paper, students of Color, teachers, schools, issues with educational policy and needed reforms?

Wish you all the best in considering some of the minor revisions and I thoroughly enjoyed reading your work. 

Reviewer 3 Report

The article is certainly interesting, but it does not apply to scientific research. It expresses the author's personal opinion about the problems, mainly of a racial nature, that have escalated during the years of the pandemic. The author's concern is supported by references to scientific studies, but there are no scientific studies in the work itself. The problems identified by the author can become sources of future research, but so far they are debatable. If your journal publishes such socially valid, but not scientific articles, then this work can be published.

Round 2

Reviewer 3 Report

As I noted earlier, the article is of great interest. The additions made to the article (the "Methods" section) confirm that the author's opinion should be taken seriously. We can consider this article as a statement of scientific problems, of a fairly general nature. Therefore, the article can be published in this form.